# Silibinin Overcomes EMT-Driven Lung Cancer Resistance to New-Generation ALK Inhibitors

**DOI:** 10.3390/cancers14246101

**Published:** 2022-12-11

**Authors:** Sara Verdura, Jose Antonio Encinar, Eduard Teixidor, Antonio Segura-Carretero, Vicente Micol, Elisabet Cuyàs, Joaquim Bosch-Barrera, Javier A. Menendez

**Affiliations:** 1Metabolism and Cancer Group, Program Against Cancer Therapeutic Resistance (ProCURE), Catalan Institute of Oncology, 17005 Girona, Spain; 2Girona Biomedical Research Institute, Salt, 17190 Girona, Spain; 3Institute of Research, Development and Innovation in Biotechnology of Elche (IDiBE) and Molecular and Cell Biology Institute (IBMC), Miguel Hernández University (UMH), 03202 Elche, Spain; 4Medical Oncology, Catalan Institute of Oncology, 17007 Girona, Spain; 5Department of Analytical Chemistry, University of Granada, 18071 Granada, Spain; 6CIBEROBN (Physiopathology of Obesity and Nutrition CB12/03/30038) Carlos III Health Institute, 28029 Madrid, Spain; 7Department of Medical Sciences, Medical School University of Girona, 17071 Girona, Spain

**Keywords:** ALK, crizotinib, brigatinib, lorlatinib, silibinin, EMT, TGFβ, lung cancer

## Abstract

**Simple Summary:**

Epithelial-to-mesenchymal transition (EMT) is a cellular plasticity program that can confer invasiveness, dissemination, and therapy resistance to cancer cells. Although inhibitors of this cellular process are expected to work as good “partners” for chemotherapy, immunotherapy or targeted therapy drugs, direct targeting of the EMT phenomenon is, in most cases, pharmacologically challenging. The objective of this work was twofold: On the one hand, to determine if the mere process of EMT is sufficient to foster the resistance of lung cancer cells to various generations of ALK tyrosine kinase inhibitors (TKIs); on the other hand, to test the capacity of the natural compound silibinin to re-sensitize lung cancer cells that gained a mesenchymal phenotype to the anti-tumor activity of ALK–TKIs. Our findings show that not all ALK-aberrant lung cancer cells exhibit the same propensity to undergo an EMT process, thereby determining whether they are able to acquire multi-resistance to various ALK–TKIs. We have also discovered the ability of silibinin to decrease the hypersecretion of the EMT-driver TGFβ, to directly block, to some extent, the activity of purified TGFβ receptors, and to attenuate the activation status of the SMAD pathway in response to ALK–TKIs. Since there exist bioavailable formulations of silibinin with proven clinical activity in oncology patients, our results suggest a new therapeutic strategy that would merit exploration to prevent or reverse resistance to ALK–TKIs induced by the EMT phenomenon.

**Abstract:**

Epithelial-to-mesenchymal transition (EMT) may drive the escape of ALK-rearranged non-small-cell lung cancer (NSCLC) tumors from ALK-tyrosine kinase inhibitors (TKIs). We investigated whether first-generation ALK–TKI therapy-induced EMT promotes cross-resistance to new-generation ALK–TKIs and whether this could be circumvented by the flavonolignan silibinin, an EMT inhibitor. ALK-rearranged NSCLC cells acquiring a bona fide EMT phenotype upon chronic exposure to the first-generation ALK–TKI crizotinib exhibited increased resistance to second-generation brigatinib and were fully refractory to third-generation lorlatinib. Such cross-resistance to new-generation ALK–TKIs, which was partially recapitulated upon chronic TGFβ stimulation, was less pronounced in ALK-rearranged NSCLC cells solely acquiring a partial/hybrid E/M transition state. Silibinin overcame EMT-induced resistance to brigatinib and lorlatinib and restored their efficacy involving the transforming growth factor-beta (TGFβ)/SMAD signaling pathway. Silibinin deactivated TGFβ-regulated SMAD2/3 phosphorylation and suppressed the transcriptional activation of genes under the control of SMAD binding elements. Computational modeling studies and kinase binding assays predicted a targeted inhibitory binding of silibinin to the ATP-binding pocket of TGFβ type-1 receptor 1 (TGFBR1) and TGFBR2 but solely at the two-digit micromolar range. A secretome profiling confirmed the ability of silibinin to normalize the augmented release of TGFβ into the extracellular fluid of ALK–TKIs-resistant NSCLC cells and reduce constitutive and inducible SMAD2/3 phosphorylation occurring in the presence of ALK–TKIs. In summary, the ab initio plasticity along the EMT spectrum may explain the propensity of ALK-rearranged NSCLC cells to acquire resistance to new-generation ALK–TKIs, a phenomenon that could be abrogated by the silibinin-driven attenuation of the TGFβ/SMAD signaling axis in mesenchymal ALK-rearranged NSCLC cells.

## 1. Introduction

The identification of molecular subtypes of non-small-cell lung cancer (NSCLC) based on specific oncogenic drivers has changed the natural history of the disease. Less than 15 years have elapsed from the first identification of the anaplastic lymphoma kinase (ALK) fusion oncogene in a patient with NSCLC [1,2] to the remarkable improvement in clinical outcomes achieved by patients with ALK-rearranged NSCLC with the first-generation ALK tyrosine kinase inhibitor (ALK–TKI) crizotinib [3,4,5]. Despite this advance, however, most patients inevitably relapse due to acquired resistance, which commonly occurs via ALK-dependent on-target mechanisms mediated by the appearance of secondary mutations in the *ALK* gene [6,7,8]. This can be observed in 25–33% of patients progressing to crizotinib [9,10,11,12,13], and increases to ~50% in response to second-generation ALK–TKIs such as ceritinib (LDK378), alectinib (CH5424802), and brigatinib (AP26113) [14,15,16]. The development of more selective and potent third-generation ALK–TKIs with improved central nervous system activity, such as lorlatinib (PF-06463922), has enabled better management of patients with resistant ALK mutant forms that are common causes of resistance against first- and second-generation ALK–TKIs [17,18,19,20,21,22]. Unfortunately, there is ever-growing evidence that several ALK-independent off-target mechanisms of acquired resistance to ALK–TKIs can occur with no involvement of ALK [23,24]. 

ALK-rearranged NSCLC tumors can lose their reliance on ALK, and instead become dependent on the alternative activation of signaling axes, for example, alterations in EGFR, KRAS/MAPK, cKIT, MET, HER2/HER3, AXL and IGF-1/IGF-1R pathways, among others [12,25,26]. Epithelial-to-mesenchymal (EMT)—a cellular process during which epithelial cells acquire mesenchymal phenotypes and behavior following the downregulation of epithelial features—is now recognized as a common downstream node in which ALK-dependent and -independent mechanisms converge to drive intrinsic and acquired resistance to ALK–TKIs [27,28,29,30,31,32,33]. Indeed, not only do ALK-rearranged tumors frequently exhibit EMT traits compared with other NSCLC genotypes, but also EMT-like processes are actively involved in mediating resistance against ALK–TKIs independently of ALK mutation status [34,35]. Furthermore, ALK-resistance mutations and an EMT component can simultaneously co-exist in two different tumor cell subpopulations in patients with ALK-rearranged NSCLC who are resistant to crizotinib [10,36]. Whether the shift from epithelial to mesenchymal phenotypes should be viewed as an ALK mutation-independent, cancer cell-autonomous phenomenon that drives cross-resistance to new-generation ALK–TKIs is still under debate [10,36]. Nonetheless, the circumvention of EMT-associated resistance to ALK–TKIs to restore the sensitivity of mesenchymal-type tumor cells to ALK–TKIs, remains an unmet need of targeted drug therapy in ALK-rearranged NSCLC. 

Here, we studied whether the EMT phenomenon that drives acquired resistance to first-generation ALK–TKI therapy suffices to promote cross-resistance to new-generation ALK–TKIs and whether the known anti-EMT [37,38,39,40]/anti-TGFβ [41,42,43,44] signaling activity of the flavonolignan silibinin could be exploited to re-sensitize drug-refractory mesenchymal NSCLC cells to ALK–TKIs, and explored the mechanisms involved. We confirm that the mesenchymal phenotype generated upon a bona fide late, full EMT phenomenon induces robust cross-resistance to multiple-generation ALK–TKIs. We also describe how the capacity of silibinin to attenuate a hyperactive TGFβ/SMAD signaling axis can overcome EMT-driven resistance to multiple-generation ALK–TKIs in ALK-rearranged NSCLC cells. 

## 2. Materials and Methods

### 2.1. Materials

Crizotinib was kindly provided by Pfizer. Brigatinib (AP26113; Cat. #S8229) and lorlatinib (PF-6463922; Cat. #S7536) were purchased from Selleckchem (Houston, TX, USA). Silibinin (Cat. #S0417) was purchased from Sigma-Aldrich (Madrid, Spain). All reagents were dissolved in sterile dimethylsulfoxide (DMSO) to prepare 10 mmol/L stock solutions, which were stored in aliquots at −20 °C until use. Working concentrations were diluted in culture medium prior to each experiment.

Antibodies against E-cadherin (#3195), SMAD2/3 (#3102) and phospho-SMAD2 (Ser465/467)/SMAD3 (Ser423/425) (#9510) were purchased from Cell Signaling Technology (Danvers, MA, USA). Antibodies against GADPH (#60004-1-Ig) and β-actin (#66009-1-Ig) were purchased from Proteintech Group, Inc (Rosemont, IL, USA). Antibodies against vimentin (#V6630) and SNAIL (#MA5-14801) were purchased from Sigma-Aldrich and ThermoFisher Scientific Inc. (Waltham, MA, USA), respectively. 

The Applied Biosystems^TM^ TaqMan^TM^ Array Human TGFβ Pathway 96-well Plate (Cat. #4414097) was purchased from Applied Biosystems (Foster City, CA, USA). RayBio^®^ C-Series Human TGFβ Array C2 (Cat. #AAH-TGFB-2-2) was purchased from RayBiotech, Inc. (Norcross, GA, USA). 

### 2.2. Cell Lines

The establishment of crizotinib resistance in H2228 cells (H2228/CR) and H3122 cells (H3122/CR) by incremental and continuous exposure to crizotinib has been described [27,45]. In order to assess the stability of acquired resistance in H2228/CR and H3122/CR cell lines, sensitivity to crizotinib was assessed after freezing and thawing as well as following drug withdrawal as previously described [46]. To generate transdifferentiated H2228 and H3122 cells (H2228/TD and H3122/TD, respectively), cells were repeatedly treated with TGFβ1 at 10 ng/mL every 3 days for 60 days. The cells were then aliquoted into vials and frozen. Newly thawed TD cells were used for up to 30 days, during which time they were exposed to TGFβ1 at 5 ng/mL once weekly. For EMT marker studies, H2228/TD and H3122/TD cells were cultured in low-serum for 24 h before treatment for an additional 24 h with 10 ng/mL TGFβ1. The SBE Reporter–HEK293 cell line (Cat. #60653; BPS Bioscience, San Diego, CA, USA) was employed for monitoring the impact of silibinin on the activity of the TGFβ/SMAD signaling pathway.

### 2.3. Quantitative Real-Time Polymerase Chain Reaction (qRT-PCR) 

Total RNA extracted from cells was evaluated in technical triplicates for the abundance of *CDH1* (Hs01023894_m1), *CDH2* (Hs00983056_m1), *VIM* (Hs00185584_m1), *SNAI1* (Hs00195591_m1), *SNAI2/SLUG* (Hs00950344_m1), and *ZEB1* (Hs00232783_m1) relative to the housekeeping genes 18*s* (Hs99999901_s1) and *PPIA* (Hs99999904_m1) using an Applied Biosystems QuantStudio^TM^ Flex PCR System with an automated baseline and threshold cycle detection. The transcript abundance was calculated using the delta Ct method and presented as relative quantification (RQ) or log2 fold-change, as specified. 

### 2.4. Immunoblotting Analyses 

HEK293 cells were seeded in 6-well plates at 250,000 cells/well and allowed to grow overnight in DMEM culture media containing 10% FBS. The media were then replaced with DMEM containing 0.1% FBS with or without TGFβ1 and/or silibinin. The cells were incubated for a further 24 h, washed with ice-cold PBS, and then immediately scraped off the plate after adding 30–75 µL of 2% SDS, 1% glycerol, and 5 mmol/L Tris-HCl, pH 6.8. Protein lysates were collected in 1.5 mL microcentrifuge tubes and the samples were sonicated for 1 min (in an ice bath) with 2 s of sonication at 2-s intervals to fully lyse the cells and reduce viscosity. Protein content was determined by the Bradford protein assay (Bio-Rad, Hercules, CA, USA). Sample buffer was added and extracts were boiled for 4 min at 100 °C. Equal amounts of protein were electrophoresed on 15% SDS-PAGE gels, transferred to nitrocellulose membranes and incubated with primary antibodies as specified, followed by incubation with a horseradish peroxidase-conjugated secondary antibody and chemiluminescence detection. GADPH and β-actin were employed as protein loading controls.

### 2.5. Cell Viability Assay 

The cell viability effects of ALK–TKIs and silibinin were determined using the colorimetric MTT (3-4,5-dimethylthiazol-2-yl-2,5-diphenyl-tetrazolium bromide) reduction assay. Dose-response curves to graded concentrations of drugs were plotted as a percentage of the control cell absorbance, which was obtained from control cells containing the vehicle processed simultaneously. For each treatment, cell viability was evaluated using the following equation: (OD_570_ of the treated sample/OD_570_ of the untreated sample) × 100. Sensitivity to agents was expressed in terms of the concentrations required for a 50% (IC_50_) reduction in cell viability. Since the percentage of control absorbance was considered to be the surviving fraction of cells, the IC_50_ values were defined as the concentration of drug that produced a 50% reduction in control absorbance (by interpolation).

### 2.6. Colony Formation Assays 

Anchorage-dependent clonogenic growth assays were performed by initially seeding NSCLC cells into 12-well plates at low densities (500–1000 cells/well) and culturing in the presence or absence of graded concentrations of ALK–TKIs and/or silibinin for 7 days (without refeeding) in a humidified atmosphere with 5% CO_2_, at 37 °C. The colonies were stained with crystal violet (0.5% *w*/*v*) in 80% methanol and 37% formaldehyde. 

### 2.7. SMAD-Binding Element Reporter Assays 

SBE Reporter–HEK293 cells were seeded at 40,000 cells per well into white clear-bottom 96-well microplates in 100 μL of assay medium and incubated at 37 °C and 5% CO_2_ overnight. The next day, the medium was removed and 3-fold serial dilutions of either SB5235443 or silibinin were prepared in the assay medium without antibiotics; 50 μL of diluted SB5235443 or silibinin was added to the wells, and 50 µL of assay medium with the same concentration of DMSO without compound was added to control wells. Additionally, 55 µL of assay medium with DMSO was added to cell-free control wells (for determining background luminescence). The cells were incubated at 37 °C and 5% CO_2_ for 4–5 h. Subsequently, 5 µL of diluted human TGFβ1 in the assay medium was added to wells (final (TGFβ1) = 20 ng/mL); 5 μL of the assay medium was added to the unstimulated control wells. The cells were treated overnight, lysed and the luciferase activity was measured using the ONE-Step luciferase assay system (BPS Bioscience): 55 μL of One-Step Luciferase reagent was added per well and the plates rocked at room temperature for ~30 min. Luminescence was measured using a BioTek Synergy^TM^ 2 luminometer (BioTek Instruments, Winooski, VT, USA). 

### 2.8. Human TGFβ Array 

Total RNA was extracted from H2228 and H2228/CR cells cultured in the absence or presence of silibinin (48 h) using the Qiagen RNeasy Kit and QIAshredder columns (Qiagen, Hilden, Germany). The Applied Biosystems^TM^ TaqMan^TM^ Array Human TGFβ Pathway 96-well plate, which contained 92 assays for TGFβ-associated genes and 4 assays for candidate endogenous control genes, was processed and analyzed as per the manufacturer’s instructions using an Applied Biosystems QuantStudio^TM^ 7 Flex PCR System. The data were interpreted using web-based PCR array analysis tools, applying a false discovery rate lower than 1% (FDR1%) and a fold-change cut-off of ≥2 (*p* < 0.05). 

### 2.9. TGFβ-Related Secretome

Assays with antibody arrays for TGFβ-related proteins were carried out as per the manufacturer’s instructions. Briefly, array membranes were blocked with 5% BSA/TBS (0.01 mol/L Tris-HCl pH 7.6/0.15 mol/L NaCl) for 1 h. The membranes were then incubated with ~1 mL of conditioned media prepared from the different cell lines after normalization for equal amounts of protein. After extensive washing with TBS/0.1% *v*/*v* Tween 20 (3 times, 5 min each) and TBS (2 times, 5 min each) to remove unbound material, the membranes were incubated with a cocktail of biotin-labeled antibodies against different individual TGFβ-related proteins. The membranes were then washed and incubated with horseradish peroxidase (HRP)-conjugated streptavidin (2.5 pg/mL) for 1 h at room temperature. Unbound HRP-streptavidin was washed out with TBS/0.1% *v*/*v* Tween 20 and TBS. Chemiluminescent readings were taken using a ChemiDoc MP imaging system (Bio-Rad Laboratories, Inc., Hercules, CA, USA) and densitometric values were quantified using ImageJ software. 

### 2.10. Docking Calculations, Molecular Dynamics Simulations, and Binding Free Energy Analysis

Docking calculations, MD simulations, and MM/PBSA calculations to determine the alchemical binding free energy of silibinin A and B against the 3D crystal structures 5E8S (human TGFβR1/ALK5) and 5E8Y (human TGFβR2 in complex with staurosporine) [47] were performed using procedures described in previous works from our group [48,49,50,51,52,53]. To perform the docking studies with AutoDockVina (v1.1.2, San Diego, CA, USA), crystal structures were transformed to the PDBQT format, including the atomic charges and atom-type definitions. These preparations were performed using the AutoDock/Vina plugin with scripts from the AutoDock Tools package [54]. YASARA dynamics v19.9.17 (Vienna, Austria) was employed for all MD simulations with the AMBER14 force field. All simulation steps were run using a pre-installed macro (md_run.mcr) within the YASARA suite. Data were collected every 100 ps during 100 ns. The MM/PBSA calculations of solvation binding energy were calculated using the YASARA macro md_analyzebindenergy.mcr, with more negative values indicating instability. MM/PBSA was implemented with the YASARA macro md_analyzebindenergy.mcr to calculate the binding free energy with solvation of the ligand, complex, and free protein, as described in [48,49,50,51,52,53]. All of the figures were prepared using PyMol 2.0 software and all interactions were detected using the protein–ligand interaction profiler (PLIP) algorithm [55].

### 2.11. LanthaScreen Eu Kinase Binding Assays

To obtain 10-point titration results of the inhibitory activity of silibinin towards the ATP-dependent kinase activity of TGFβR1/ALK5 and TGFβ2R, LanthaScreen Eu kinase binding assays were outsourced to ThermoFisher Scientific using the SelectScreen^TM^ Biochemical Kinase Profiling Service. 

### 2.12. Statistical Analysis 

All observations were confirmed by at least three independent experiments performed in triplicate for each cell line and for each condition. The data are presented as mean ± SD. Two-group comparisons were performed using Student’s *t*-test for paired and unpaired values. Comparisons of means of ≥3 groups were performed by ANOVA, and the existence of individual differences, in case of significant F values with ANOVA, was tested by Scheffé’s multiple contrasts; *p*-values <0.05 and <0.005 were considered to be statistically significant (denoted as * and **, respectively). All statistical tests were two-sided.

## 3. Results

### 3.1. Acquisition of a Mesenchymal-Like Phenotype Promotes Cross-Resistance to First-, Second-, and Third-Generation ALK–TKIs in ALK-Rearranged NSCLC Cells

To explore whether acquired resistance to first-generation crizotinib might be accompanied by cross-resistance to second- and third-generation ALK–TKIs in an EMT-dependent manner, we characterized two crizotinib-resistant sublines (H2228/CR and H3122/CR) derived from the crizotinib-sensitive H2228 and H3122 NSCLC cell lines harboring the ALK variants 3a/b and 1, respectively [56]. H2228/CR and H3122/CR cells were derived by incremental and continuous exposure of parental lines to increasing concentrations of crizotinib over several months [27,45,57]. H2228/CR and H3122/CR cells lack amplification or resistance mutations in the ALK kinase domain, thus offering two idoneous models to explore the involvement of EMT as an ALK-independent, off-target resistance mechanism to new-generation ALK–TKIs (Figure 1). 

Examination of morphological and molecular features of H2228/CR and H3122/CR cells revealed that the characteristic “cobblestone” morphology of parental H2228 epithelial cells was absent in H2228/CR cells, which instead assumed an elongated morphology with evident disruption of tight cell-cell contacts and a notably lower refringent aspect (Figure 1A). By contrast, H3122/CR cells failed to fully phenocopy the mesenchymal-like morphology of H2228/CR cells as they acquired a more marked spindle-shaped morphology and retained numerous cell-cell contacts and a refringent aspect (Figure 1A). Quantitative real-time PCR (qRT-PCR) analyses revealed that H2228 cells exhibited more EMT-like traits than H3122 cells in terms of mesenchymal markers such as vimentin (VIM) (Figure 1B). Crizotinib resistance in H2228/CR cells was accompanied by the acquirement of a bona fide EMT program involving a marked transcriptional down-regulation of the epithelial marker E-cadherin (CDH1) and activation of EMT-driven transcription factors and EMT-related markers (SNAI1, VIM) (Figure 1B). Crizotinib resistance in H3122/CR cells was also accompanied by a notable gain in mesenchymal gene expression including the mesenchymal N-cadherin (CDH2) and VIM, but CDH1 expression was retained (Figure 1B). 

MTT-based viability assays revealed notably higher half-maximal inhibitory concentration (IC_50_) values to crizotinib in H2228 cells than in H3122 cells, confirming that NSCLC cells with the variant 3a/b have an inferior response to ALK–TKIs and more aggressive behavior that those with variant 1 (Appendix A; Figure 1C) [58,59,60,61]. H2228/CR cells, which were ~5-fold more resistant to crizotinib than parental H2228 cells, showed substantial cross-resistance to the second-generation ALK–TKI brigatinib (~8-fold increase in IC_50_) and were largely unresponsive to the cytotoxic effects of the third-generation ALK–TKI lorlatinib. Indeed, a >80-fold higher concentration of lorlatinib was necessary to obtain an IC_50_ in H2228/CR cells compared with parental H2228 parental cells (Appendix A; Figure 1C). Although H3122/CR cells exhibited a similar cross-resistance to crizotinib, brigatinib, and lorlatinib (between ~4- and 6-fold), the IC_50_ values of ALK–TKIs against H3122/CR cells were substantially lower than those for H2228/CR cells (>3000-fold for lorlatinib; Appendix A; Figure 1C). 

Overall, these findings strongly suggest that when a bona fide, full mesenchymal phenotype develops upon chronic exposure of intrinsically aggressive variant 3a/b-harboring ALK-rearranged NSCLC cells to a first-generation ALK–TKI (crizotinib), those cells are no longer responsive to second and third-generation ALK–TKIs. This cross-resistance phenotype is less pronounced when intrinsically sensitive variant 1-harboring ALK-rearranged NSCLC cells acquire a partial E/M transition state.

### 3.2. Silibinin Re-Sensitizes Mesenchymal NSCLC Cells to ALK–TKIs

We next examined the ability of the flavonolignan silibinin to re-sensitize mesenchymal cells to ALK–TKIs. H3122/CR cells showed a notably improved sensitivity to crizotinib (~3-fold), brigatinib (~6-fold), and lorlatinib (~4-fold) when MTT-based IC_50_ values were re-calculated in the presence of an optimal concentration of silibinin (100 μmol/L) (Appendix A; Figure 2A). Although silibinin co-exposure also decreased the IC_50_ values of ALK–TKIs against H3122/CR cells, such sensitizing activity could be attributed to silibinin toxicity as single agent (Appendix A; Figure 2B). 

To further examine the sensitizing effects of silibinin on EMT-driven cross-resistance to ALK–TKIs, we performed long-term colony formation assays using doses of ALK–TKIs optimized to maximally discriminate between cell growth in monotherapy and combination therapy with silibinin (75 μmol/L). The combination of ALK–TKIs and silibinin was markedly more effective than ALK–TKIs or silibinin used in monotherapy in attenuating the colony formation potential of mesenchymal-like H2228/CR cells (Figure 2C, left panels). Co-treatment with silibinin re-sensitized non-mesenchymal H3122/CR cells to crizotinib; however, less evident changes were observed when combining silibinin with sub-optimal concentrations of brigatinib and lorlatinib, which remained highly active against H3122/CR cells even at nanomolar concentrations (Figure 2C, right panels).

Because exacerbated TGFβ1 signaling has been shown to drive the EMT-like phenotype in H2228/CR cells [27], we evaluated the ability of silibinin to modulate ALK–TKI activity in a transdifferentiated (TD) cell model established by chronic exposure of H2228 cells to TGFβ1 (Appendix A). qRT-PCR analyses confirmed that long-term treatment of H2228 cells with TGFβ (10 ng/mL) was sufficient by itself to induce EMT, as characterized by the acquisition of mesenchymal-like morphological traits equivalent to those found in H2228/CR cells, including the up-regulation of the EMT markers SNAI1, SLUG, VIM, and ZEB1 and the marked downregulation of CDH1 expression (Appendix A). Additionally, H2228/TD cells exhibited a cross-resistant phenotype to multiple-generation ALK–TKIs, which was particularly striking for the third-generation lorlatinib (>9-fold increase in the IC_50_ value of H2228/TD cells compared with parental H2228 cells; Appendix A). H3122/TD failed to acquire a bona fide activation of the EMT transcriptional program after TGFβ1 stimulation, with the exception of a notable up-regulation of VIM (Appendix A). Chronic TGFβ stimulation failed to promote acquired resistance to crizotinib but significantly augmented the IC_50_ values of brigatinib and lorlatinib (~5-fold increase in the case of the third-generation ALK–TKI lorlatinib; Appendix A). Silibinin treatment significantly reduced the IC_50_ values of ALK–TKIs against H2228/TD and H3122/TD cells (Appendix A). 

Altogether, these findings strongly suggest that silibinin re-sensitizes mesenchymal NSCLC cells to ALK–TKIs, at least in part, by targeting the EMT-driving TGFβ signaling.

### 3.3. Silibinin Suppresses the TGFβ/SMAD Signaling Pathway

Given our findings thus far, we evaluated whether the acquisition of the mesenchymal phenotype in ALK–TKI-refractory H2228/CR cells involved changes in TGFβ/SMAD signaling [62,63,64]. Immunoblotting analysis confirmed an increase in total SMAD3 expression in H2228/CR cells concomitant with the constitutive hyperactivation of regulatory SMADs (SMAD2 and SMAD3; Figure 3A), which was largely phenocopied by chronic TGFβ1 stimulation in H2228/TD cells (Appendix A). Activation of SMAD signaling in H2228/CR cells was accompanied by the conspicuous loss of E-cadherin expression, a slight increase in the abundance of vimentin, and a marked accumulation of the EMT-inducible transcription factor SNAIL (Figure 3A). By contrast, no significant changes were observed in the phosphorylation status of regulatory SMADs in H3122/CR cells, which fully retained the expression of E-cadherin along with a significant up-regulation of vimentin but no induction of SNAIL expression (Figure 3A). Chronic stimulation with TGFβ1 in H3122/TD cells, however, notably promoted both vimentin and SNAIL expression (Appendix A).

We next tested whether TGFβ/SMAD signaling could be targeted by silibinin using the TGFβ/SMAD Signaling Pathway SBE Reporter-HEK293 cell line, a stable transfected HEK293 cell line expressing the Renilla luciferase gene under the transcriptional control of synthetic SMAD binding elements (SBE) (Figure 3B). When SBE activity was measured in SBE-HEK293 cells stimulated with TGFβ1 in the absence or presence of graded concentrations of silibinin for 24 h, we observed a dose-dependent inhibition of TGFβ1-induced SBE activity with an IC_50_ value of ~25 μmol/L (Figure 3B). To confirm that silibinin can shut down the activation of SMADs as intracellular signaling mediators transducing TGFβ1 extracellular signals to the nucleus, SBE Reporter-HEK293 cells were treated with TGFβ1 in the absence or presence of either silibinin or SB431542, a potent inhibitor of intracellular TGFβ signaling. The results showed a time-dependent increase in the levels of phospho-SMAD2/3 upon TGFβ1 treatment, whereas silibinin co-treatment largely mimicked SB431542 in preventing TGFβ1-induced SMAD2/3 phosphorylation (Figure 3B). 

We next evaluated how silibinin treatment might impact the transcriptional expression of TGFβ/SMAD-responsive genes in ALK–TKI-responsive H2228 and ALK TKI-refractory H2228/CR cells using the Applied Biosystems^TM^ TaqMan^TM^ Array Human TGFβ Pathway panel (Figure 3C). Of the 92 assays for ligands, receptors, and mediators of the TGFβ/BMP superfamily, the analyses revealed 7 genes exclusively down-regulated in parental H2228 cells (GDF5, SOX4, ACVRL1, INHBA, BMP2, TSC22D1, TGFB1 ∣ 1), 9 genes commonly down-regulated in H2228 and H2228/CR cells (SMAD1, BMP4, TGFB3, NOG, COL1A1, ID1, TNFSF10, BMPR1A, RUNX1), and 11 genes exclusively regulated in H2228/CR cells (THBS1, TGFBR2, TGFB2, TGFB1, BMP6, BMP1, COL1A2, SMAD3, ATF4, FST, SMURF1) (Figure 3C). 

### 3.4. Silibinin Is Predicted to Directly Inhibit the Kinase Activity of TGFβR1/2 

As the complex EMT-promoting function of TGFβ depends on the activation of the highly conserved single transmembrane serine/threonine kinases type 1 (TGFβR1 or ALK5) and type 2 (TGFβR2) receptors, we explored the possibility that silibinin might directly inhibit TGFβR1/2 kinase activity. 

To initially test a putative interaction or binding of silibinin with TGFβ receptors, we computationally docked silibinin into the ATP/ligand binding pocket of TGFβR1/ALK5 and TGFβR2 (Figure 3D). As silibinin is almost a 1:1 mixture of the diastereomers A and B, we performed classical molecular docking studies of silibinin A and B against the 3D crystal structures 5E8S (human TGFβR1/ALK5) and 5E8Y (human TGFβR2 in complex with staurosporine) [46]. The resulting binding energies with the docking simulations of TGFβR1/ALK5 (−9.753 [A] and −10.73 [B] kcal/mol) and TGFβR2 (−9.551 [A] and −12.02 [B] kcal/mol) were marginally superior for the diastereomer B against TGFβR2. To better understand the predicted tendencies, we performed molecular dynamics (MD) simulations for each of the TGFβR1/2-silibinin A/B complexes (Figure 4). 

The MD approach considers the protein flexibility at the target-binding site during the molecular recognition process, thereby allowing confirmation of the kinetic stability and validation of the binding poses obtained by docking. The TGFβR1/2 protein backbone root mean square deviation (RMSD) plots of the silibinin heavy atoms, measured after superimposing TGFβR1/2 onto its (apo) reference structure during MD simulations, were prepared in parallel. This approach is summarized in Figure 4A and detects the following: the best poses of silibinin A and silibinin B coupled to the catalytic cavities of TGFβR1/2 before (0 ns) and after (100 ns) the MD simulation, the time evolution of RMSD relative to the initial structure of TGFβR1/2 in the absence and presence of silibinin A/B, the binding free energy calculations under the molecular mechanics Poisson–Boltzmann surface area (MM/PBSA) approximation from the entire MD simulation trajectory of 100 ns (or last 30 ns), and the identification of amino acid residues participating in the silibinin A/B-TGFβR1/2 binding pocket. Close inspection of the different conformations revealed that silibinin A was not predicted to interact with TGFβR1 His-283 (or its equivalent in TGFβR2 His-328) or TGFβR1 Asp-281 (or its equivalent in TGFβR2 Ala-326), which are two key residues in the hinge region of TGFβRs critically involved in the binding of selective TGFβR1 and pan-TGFβR1/2 inhibitors [46]. Silibinin A was predicted to stably interact throughout the entire MD simulation with TGFβR1 Lys-232, the third key residue in the hinge region, as well as with TGFβR1 Ile-211 and Val-219, two residues establishing non-polar contacts with pan-TGFβR1/2 and selective TGFβR1 inhibitors. By contrast, silibinin B was predicted to interact with the key residues TGFβR2 His-328/Ala-326 as well as with Val-250, Val-258, and Leu-386, three residues establishing non-polar contacts with pan-TGFβR1/2 inhibitors. Moreover, silibinin B was predicted to stably interact with Lys-277, a crucial residue located at the ATP-binding site whose mutation destroys the kinase and signaling activities of TGFβR2 [65]. 

We used LanthaScreen Eu kinase binding assays to test whether silibinin could function as a TGFβR1/2 kinase inhibitor. This assay monitors the displacement of a labeled “tracer” (Alexa Fluor^TM^ conjugate) from a protein (in our case TGFβRs) by a putative inhibitor, which is detected as a loss of fluorescence resonance energy transfer (FRET) (Figure 4B, left panel). Dose-response curves showed that the emission ratio was dose-dependently decreased by silibinin with IC_50_ values against TGFβR1/ALK5 and TGFβR2 of 70 and 56 μmol/L, respectively (Figure 4B, right panel). 

The computational modeling and in vitro enzymatic analyses altogether indicate that silibinin could bind the ATP-binding sites to operate as a direct inhibitor of the TGFβR1/2 kinase activities but solely at the two-digit micromolar range.

### 3.5. Silibinin Normalizes TGFβ Oversecretion and SMAD2/3 Hyperactivation in ALK–TKI-Resistant NSCLC Cells

We explored whether silibinin treatment might impact both the secretome for proteins linked to the TGFβ signaling pathway and the activation of SMAD2/3 in ALK–TKI-resistant NSCLC cells. We took advantage of the RayBio^®^ C-Series Human TGFβ Array C2 (RayBiotech, Inc., Norcross, GA, USA), which simultaneously detects twenty-five TGFβ signaling-related proteins (Figure 5). As expected, we noticed that TGFβ1 was notably elevated in the culture supernatant of H2228/CR cells compared with H2228 parental cells [27]. Although less markedly, higher levels of TGFβ1 were detected in the in the culture supernatant of H3122/CR cells compared with H3122 parental cells. Silibinin treatment reverted the oversecretion of TGFβ1 in H2228/CR and H3122/CR cells back to the baseline levels found in H2228 and H3122 parental cells (Figure 5). The secretion levels of the divergent member of the TGFβ superfamily GDF15 were found to be drastically decreased in culture supernatants from H2228/CR cells. Moreover, whereas silibinin treatment further augmented baseline GDF15 secretion in H2228 parental cells, it partially recovered the extremely low levels of GDF15 in H2228/CR cells. 

Immunoblotting procedures confirmed that silibinin treatment partially but significantly alleviated the constitutive hyperactivation of SMAD2/3 in H2228/CR cells irrespective of the presence of ALK–TKIs (Figure 5). Moreover, ALK–TKIs were found to promote a marked phosphorylation of SMAD2/3 in H3122 and H3122/CR cells (e.g., brigatinib), an activating effect that was largely prevented in the presence of silibinin (Figure 5). 

## 4. Discussion

The mesenchymal phenotype induced by EMT appears to be an independent resistance mechanism to the first-generation ALK–TKI crizotinib in patients with ALK-rearranged NSCLC [10,36]. If this also occurs in relation to second- and third-generation ALK–TKIs with activity against crizotinib-resistant ALK mutations, the EMT phenomenon could significantly compromise the possible use of next-generation ALK–TKIs as first-line treatment in ALK-rearranged NSCLC. We show that the acquisition of a mesenchymal phenotype by ALK-rearranged NSCLC cells following chronic exposure to crizotinib or to TGFβ stimulation increases resistance to the second-generation ALK–TKI brigatinib and promotes full refractoriness to the third-generation ALK–TKI lorlatinib. Our findings also identify the flavonolignan silibinin as a potential candidate for treating EMT-driven cross-resistance to new-generation ALK–TKIs.

There is evidence from cell line-based experimental models and from in vivo profiling of post-treatment biopsy specimens from ALK–TKI-resistant tumors strongly supporting EMT as a central off-target mechanism of acquired resistance to ALK–TKIs without the involvement of *ALK* mutations [10,27,36]. Indeed, sustained ALK activity driven by different ALK rearrangements induces an EMT signature in NSCLC but with a noteworthy degree of heterogeneity [35]. ALK-rearranged NSCLC cells exhibiting an EMT-like signature are intrinsically less sensitive to ALK–TKIs than equivalent cells with an epithelial-like signature [66]. Moreover, the acquisition of resistance to ALK–TKIs associates with an EMT phenotype that can be secondary to activation of TGFβ signaling induced by hypoxia or by yet-to-be-defined mechanisms [27,67,68,69]. Lastly, although ALK-resistant mutations and mesenchymal tumor cells can coexist in a single crizotinib-resistant lesion, the ALK-resistant mutation is largely restricted to epithelial-type tumor cells, whereas tumor cells with the mesenchymal phenotype can exhibit cross-resistance to crizotinib and new-generation ALK–TKIs, including alectinib, ceritinib, and lorlatinib [36]. We provide evidence that ALK-rearranged NSCLC cells gaining a bona fide mesenchymal phenotype caused by a late, full EMT upon chronic exposure to crizotinib, but not those acquiring only a partial/hybrid E/M transition state, exhibit cross-resistance to multiple-generation ALK–TKIs (Figure 6). Our data strongly support a molecular scenario wherein the plasticity along the EMT spectrum determines the propensity of ALK-rearranged NSCLC cells to exhibit cross-resistance to multiple-generation ALK–TKIs. Accordingly, the more epithelial an ALK-rearranged NSCLC cell population is, the lower the capacity to acquire a mesenchymal phenotype refractory to new-generation ALK–TKIs, and vice versa. As cellular heterogeneity along this spectrum is a paramount feature in most tumors including ALK-rearranged NSCLC, forthcoming studies should evaluate whether the utilization of the so-called EMT scores, which have been developed based on pan-cancer signatures of EMT identified from preclinical and/or clinical data [70,71,72,73,74], in a primary/metastatic tumor can be used to predict resistance to ALK–TKIs. 

A pioneering study on the functional landscape of resistance to ALK inhibition in lung cancer proposed several possible agents (including inhibitors of EGFR, HER2/HER3, or PKC) that might be combined with ALK inhibitors to overcome or delay a range of resistance mechanisms in ALK-rearranged NSCLC cells (H3122) with marked sensitivity to ALK–TKIs [78]. Nevertheless, novel therapeutic strategies capable of circumventing EMT underpinning short-lived responses to various cytotoxic and targeted drugs including multiple-generation ALK–TKIs remain an unmet clinical need in ALK-rearranged NSCLC [79,80,81,82,83,84]. The flavonolignan silibinin, the bioactive principle of the silymarin extract isolated from the dried fruits of the milk thistle (*Silybum marianum*) [85,86,87,88], has been shown to exert anti-tumor activities, at least in part, by targeting EMT-related molecular traits in cancer cells. Its ability to concurrently prevent the loss of epithelial markers (E-cadherin) and activate proteins associated with the mesenchymal phenotype (vimentin, N-cadherin, CD44) was previously ascribed to its regulatory effects on major EMT transcriptional regulators, including the transcription factors SNAIL, SLUG, and ZEB and the microRNA miR-21/miR-200c [37,38,39,40,57,89,90,91,92]. Additionally, silibinin was shown to inhibit fibrotic responses in several tissues via suppression of TGFβ1/SMAD2/3 signaling [42,44,93]. We confirm here the ability of silibinin to control TGFβ/SMAD signaling, as demonstrated by the deactivation of SMAD2, the prevention of SBE-controlled transcriptional responses, and the transcriptional down-regulation of TGFβ-associated genes. TGFβ signaling is initiated by ligand binding to TGFβR2 (TβRII, TGFBR2), a transmembrane receptor with intracellular serine/threonine kinase activity [62,63,64]. Ligand binding leads to dimerization and autophosphorylation of TGFβR2, which then binds and stimulates the serine/threonine kinase activity of TGFβR1/ALK5. In turn, TGFBR1/ALK5 phosphorylates the cytoplasmic signaling proteins SMAD2 and SMAD3, which associate with SMAD4 to translocate into the nucleus as a multiprotein complex that stimulates the transcription of TGFβ-responsive genes. Our study might add a missing piece to the mechanistic puzzle of the anti-EMT activity of silibinin by revealing that it binds the ATP-binding domain of TGFβR kinases, inhibiting their ATP kinase activity and blocking downstream signaling cascades. In silico, silibinin is predicted to interact with the catalytic site of TGFβR1/ALK5 and TGFβR2, showing shared but mostly distinct contacts to pan- and selective TGFβR inhibitors [46]. These findings confirm not only that flavonolignans such as silibinin should be viewed as specific ligands of biological targets according to the “lock-and-key” concept, but also that the two silibinin diastereomers A and B might behave differently in terms of their biological activity as optically pure components against TGFβRs [94,95]. 

Beyond underscoring a possible role for stereochemistry in determining the inhibitory potency of silibinin against TGFβRs, we failed to observe a good correlation between the timeline representation of MM/PBSA binding energies of the silibinin diastereomers and the experimental inhibitory activities of the diastereomeric mixture of silibinin A/B using the LanthaScreen^TM^ Eu Kinase Binding Assay, which is established on the binding and displacement of an ATP-competitive kinase inhibitor scaffold to the TGFβR1 and 2 kinases. As the tracers are based on ATP-competitive kinase inhibitors, they are suitable for detection of any compound that binds the ATP site, including those that bind to both the ATP site and a second “allosteric” site. Our in silico versus experimental data highlight the importance of the use of the respective optically pure components of the silibinin diastereomeric pair to molecularly understand (and therapeutically develop) the anti-TGFβR inhibitory activity of silibinin. Whether the discrepancy between in silico predictions and the observed dose-response curves of silibinin against TGFβR1 and 2 at micromolar concentrations in vitro involves the presence of various inhibitor sites at the kinases or other enzyme-inhibitor parameters (e.g., enzyme concentration >> K_d_ value of silibinin) deserves careful consideration in the further development of silibinin as an anti-TGFβR/SMAD signaling therapeutic [96]. More importantly, one should acknowledge that the ability of silibinin to function as a direct inhibitor of the TGFβR1/2 kinase activities took place at the two-digit micromolar range, which makes a direct and unique mechanistic involvement of the TGFβR1/2 kinase activities in the ALK–TKIs sensitizing activity of silibinin to some extent improbable. A secretome profiling confirmed the ability of silibinin to normalize the augmented release of TGFβ into the extracellular fluid of ALK–TKIs-resistant NSCLC cells while significantly reducing constitutive and inducible SMAD2/3 phosphorylation in the presence of ALK–TKIs. The ability of silibinin to normalize the enhanced expression and augmented secretion of the EMT-driving factor TGFβ1 into the extracellular milieu might rather explain, at least in part, its ability to attenuate the TGFβ/SMAD signaling axis in ALK–TKIs-resistant NSCLC cells. Nonetheless, as ALK–TKI resistance based on EMT-like phenomena has cross-sensitivity to inhibitors of the Hsp90 chaperone such as ganetespib, 17-AGG, 17-DMAG, and NVP-AUY922 [22,27,69,97,98,99], we cannot exclude the possibility that the reported capacity of silibinin as a novobiocin-like Hsp90 inhibitor could promote the degradation of client proteins, including not only mutant ALK but also TGFβRs in mesenchymal ALK-rearranged cells with acquired resistance to ALK–TKIs [100,101]. Our previous experience with water-soluble, bioavailable formulations of silibinin demonstrated a complete abrogation of tumor growth in xenograft models of EMT-driven resistance to EGFR TKIs [39,40]. Forthcoming studies should take up the challenge of confirming if clinically relevant formulations of silibinin (e.g., silibinin complexed with the amino-sugar meglumine; silibinin-phosphatidylcholine, the phytolipid delivery system Siliphos; and Eurosil^85^/Euromed, a milk thistle extract that is the active component of the nutraceutical Legasil with enhanced bioavailability [102]) could similarly abrogate the ALK–TKIs-refractory tumor growth in vivo. 

## 5. Conclusions

The ab initio plasticity along the EMT spectrum should be viewed as a key determinant of the propensity of ALK-rearranged NSCLC cells to acquire resistance to new-generation ALK–TKIs. EMT-driven NSCLC cross-resistance can be abrogated by silibinin, which directly inhibits TGFβR kinase activity and blocks the SMAD signaling cascade in mesenchymal ALK-rearranged NSCLC cells. As EMT is an increasingly recognized driver of innate and acquired resistance to various cytotoxic and targeted drugs, clinically-relevant bioavailable formulations of silibinin with proven anti-cancer activity [103,104,105] could be explored as cost-effective and feasible approaches for patients with NSCLC resistant to ALK–TKIs. 

## Figures and Tables

**Figure 1 cancers-14-06101-f001:**
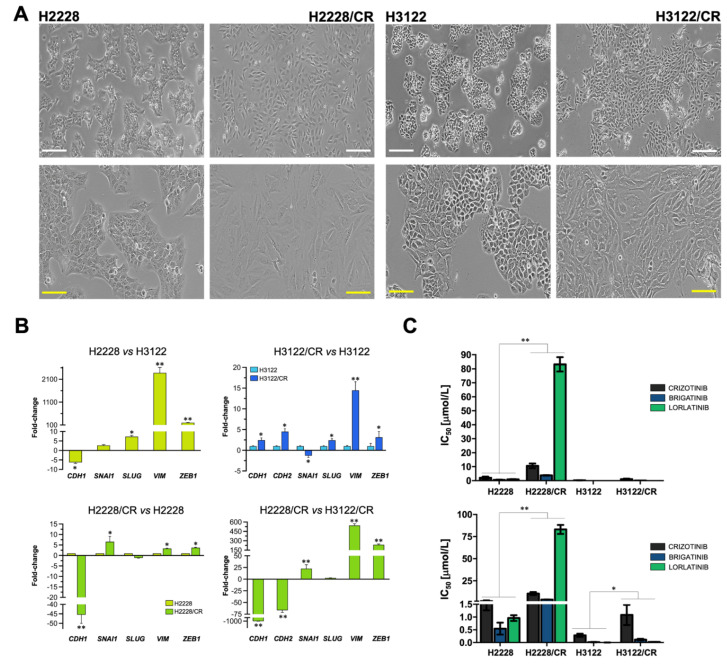
EMT-related traits in ALK-rearranged NSCLC cells with acquired cross-resistance to multiple-generation ALK–TKIs. (**A**) Representative phase contrast microphotographs of H2228/H2228CR and H3122/H3122CR ALK-rearranged NSCLC cell line pairs. CR: crizotinib resistance; Scale bar, 100 μm. (**B**) The transcript abundance of *CDH1*, *CDH2*, *SNAI1*, *SLUG*, *VIM*, and *ZEB1* was calculated using the ΔC_t_ method and presented as fold-change in H2228/H2228CR and H3122/H3122CR cells; * *p* < 0.05 and ** *p* < 0.005, statistically significant differences. (**C**) Bar graphs showing the MTT-based IC_50_ values of crizotinib, brigatinib, and lorlatinib for H2228/H2228CR and H3122/H3122CR cells. The results are presented as the means (*columns*) ± S.D (*bars*) (*n* = 5, in triplicate). * *p* < 0.05 and ** *p* < 0.005, statistically significant differences.

**Figure 2 cancers-14-06101-f002:**
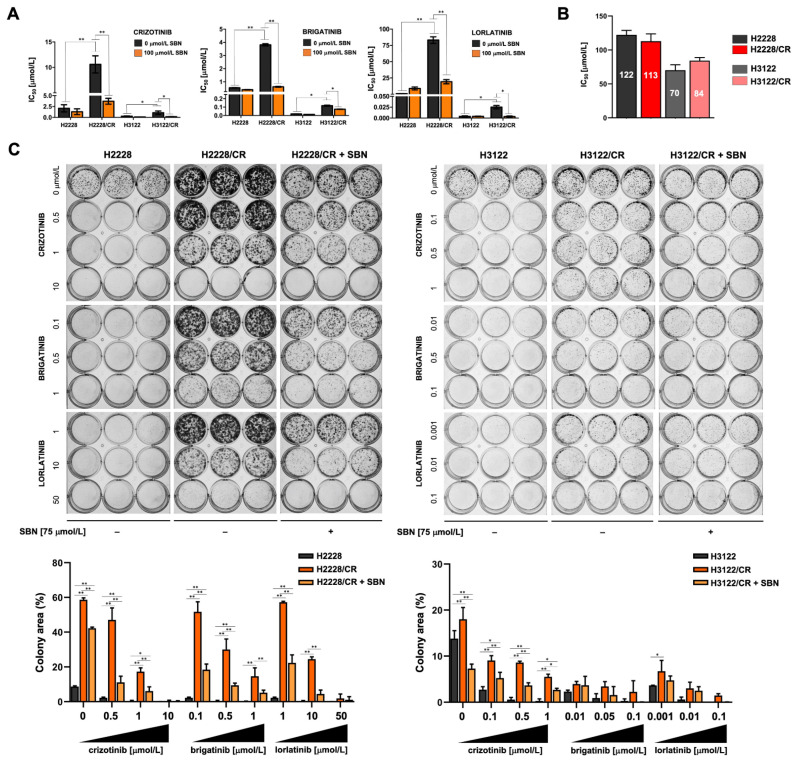
Sensitizing effects of silibinin against EMT-related acquisition of cross-resistance to multiple-generation ALK–TKIs. (**A**) Bar graphs showing the MTT-based IC_50_ values of crizotinib, brigatinib, and lorlatinib for H2228/H2228CR and H3122/H3122CR cells calculated in the absence or presence of 100 μmol/L of silibinin. (**B**) Bar graphs showing the MTT-based IC_50_ values of silibinin in H2228/H2228CR and H3122/H3122CR cells. The results in A and B are presented as the means (*columns*) ± S.D (*bars*) (*n* = 5, in triplicate). * *p* < 0.05 and ** *p* < 0.005, statistically significant differences; n.s. not significant. (**C**) Top: Representative images of clonogenic survival analyses (7 days) of H2228/H2228CR (left) and H3122/H3122CR cells (right) in response to graded concentrations of ALK–TKIs in the absence or presence of 75 μmol/L silibinin. Bottom: Colony area (%) was calculated using the ImageJ plugin “ColonyArea”. The results are presented as the means (*columns*) ± S.D (*bars*) (*n* = 3, in triplicate). * *p* < 0.05 and ** *p* < 0.005, statistically significant differences; n.s. not significant.

**Figure 3 cancers-14-06101-f003:**
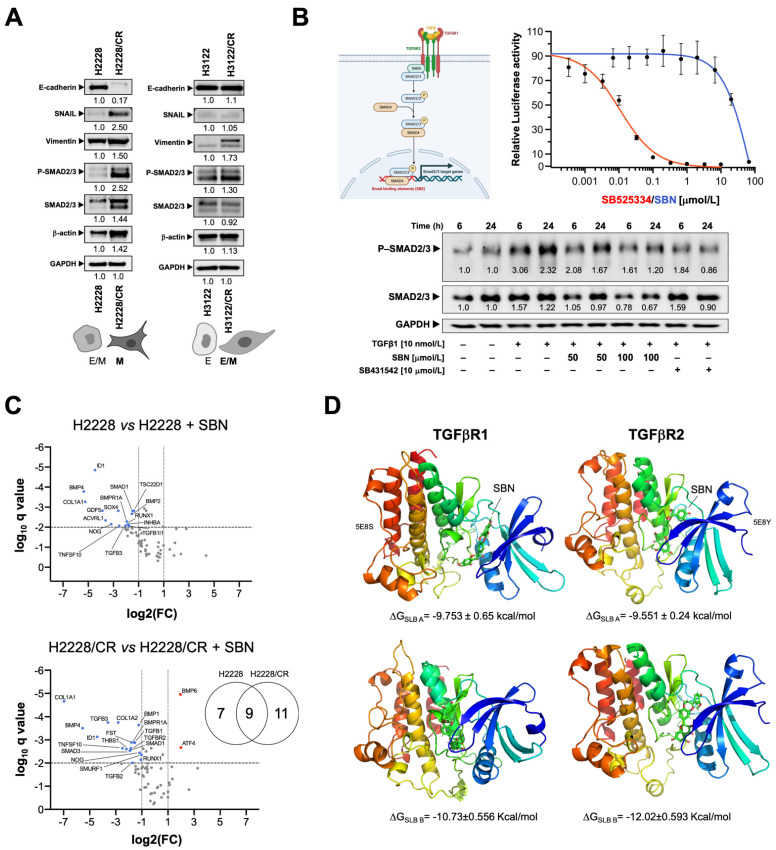
Targeted effects of silibinin against the TGFβ/TGFβR/SMAD signaling pathway. (**A**) Expression levels of E-cadherin, SNAIL, vimentin, phospho-SMAD2/3, total SMAD2/3 were detected by immunoblotting in H2228/H2228CR and H3122/H3122CR cells using specific antibodies. The intensity of the bands was measured using the ImageJ software. Fold-change of each protein relative to parental cells was calculated using GAPDH as a loading control. The figure shows representative immunoblots of multiple (*n* ≥ 5) independent experiments. E: Epithelial; M: Mesenchymal. (**B**) Top: Relative luciferase activity using SBE Reporter–HEK293 cells pre-incubated during 4–5 h with graded concentrations of SB525334 and silibinin before stimulation with TGFβ1. Bottom: Expression levels of phospho-SMAD2/3 and total SMAD2/3 were detected by immunoblotting in HEK293 cells stimulated with TGFβ1 (0, 6, and 24 h) in the absence/presence of either silibinin or SB431542 using specific antibodies. The intensity of the bands was measured using the ImageJ software. Fold-change of each protein relative to untreated samples was calculated using GAPDH as a loading control. The figure shows representative immunoblots of multiple (*n* ≥ 3) independent experiments. (**C**) Volcano plots of the results from analyses of the Applied Biosystems^TM^ TaqMan^TM^ Array Human TGFβ Pathway in H2228/H2228CR cells cultured in the absence/presence of silibinin (100 μmol/L) for 48 h. Each dot represents a transcript with its corresponding mean Log2 fold-change (FC) (*x* axis) and Benjamini–Hochberg corrected *p*-value (−log10, *y* axis). Colored dots illustrate differential lipid species, using a cutoff of *p* < 0.05 and log2FC > 1 or < 1. (**D**) The figure depicts the backbone of the overall crystal structure of TGFβR1 (5E8S) and TGFβR2 (5E8Y) with rainbow colors showing the best docked poses of silibinin A and silibinin B at the catalytic site. The uncropped western blot figures were presented in Appendix A.

**Figure 4 cancers-14-06101-f004:**
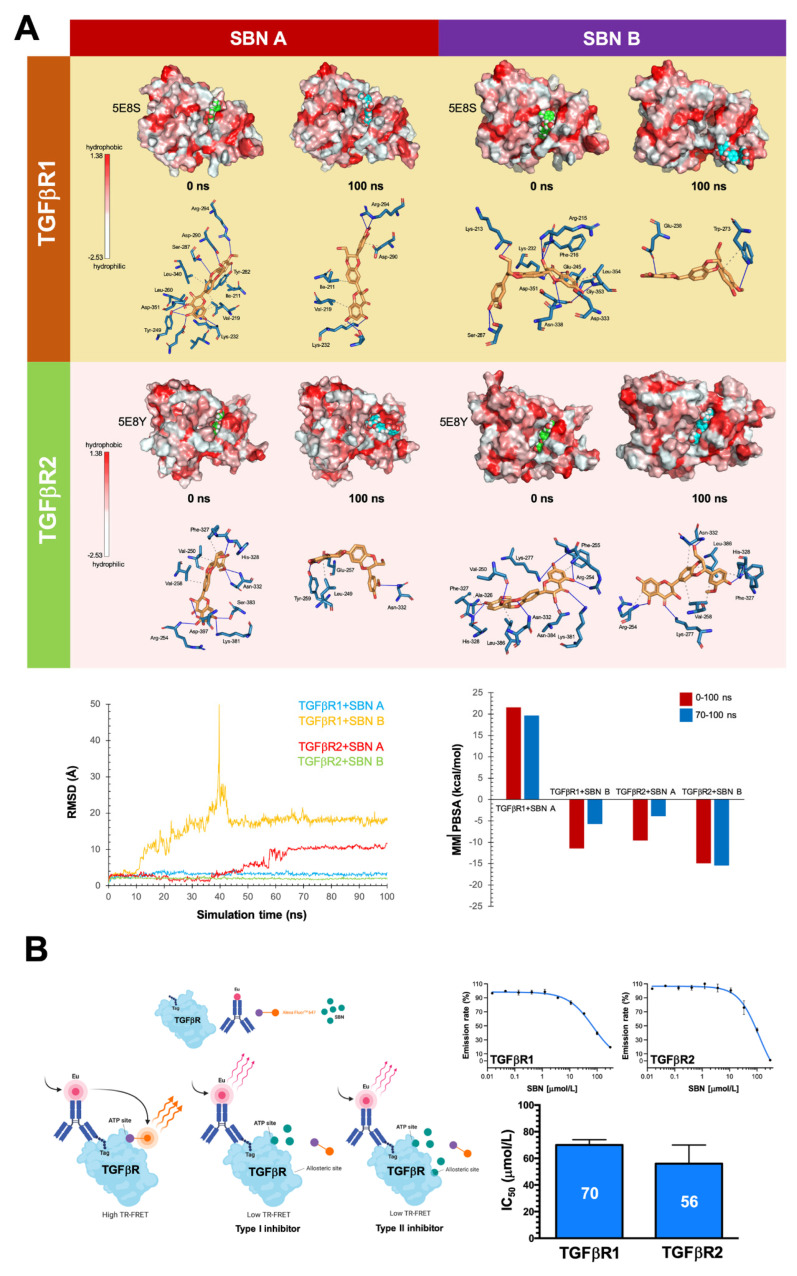
In silico prediction and in vitro verification of silibinin as a weak, direct inhibitor of TGFβRs 1 and 2. (**A**) Top: The best poses of silibinin A and silibinin B coupled to the catalytic site of TGFβR1 (5E8S) and TGFβR2 (5E8Y) before (0 ns) and after (100 ns) the molecular dynamics (MD) simulations are shown. The protein is represented as a function of the hydrophobicity of its surface amino acids, and the Na^+^ and Cl^−^ ions have been eliminated to facilitate visualization. Each inset shows the detailed interactions of the participating amino acids involved and the type of interaction (hydrogen bonds, hydrophilic interactions, salt bridges, Π-stacking, etc). Bottom: The root means square deviation (RMSD, Å) of the heavy atoms of silibinin A and silibinin B over the simulation time, measured after superposing the protein onto its reference structure, and the molecular mechanics Poisson–Boltzmann surface area (MM/PBSA) binding energy analyses calculated from the entire trajectory of the 100 ns (or last 30 ns) MD simulation, are shown. (**B**) Top: Dose-response curves of LanthaScreen Eu TGFβR1 and TGFβR2 kinase binding assays showing dose-dependent decreases in emission ratios induced by graded concentrations of silibinin. Bottom: Bar graphs showing the IC_50_ values of silibinin for the ATP-dependent activity of TGFβR1 and TGFβR2. The results are presented as the means (*columns*) ± S.D (*bars*). All experiments were carried out two times in duplicate to assess reproducibility.

**Figure 5 cancers-14-06101-f005:**
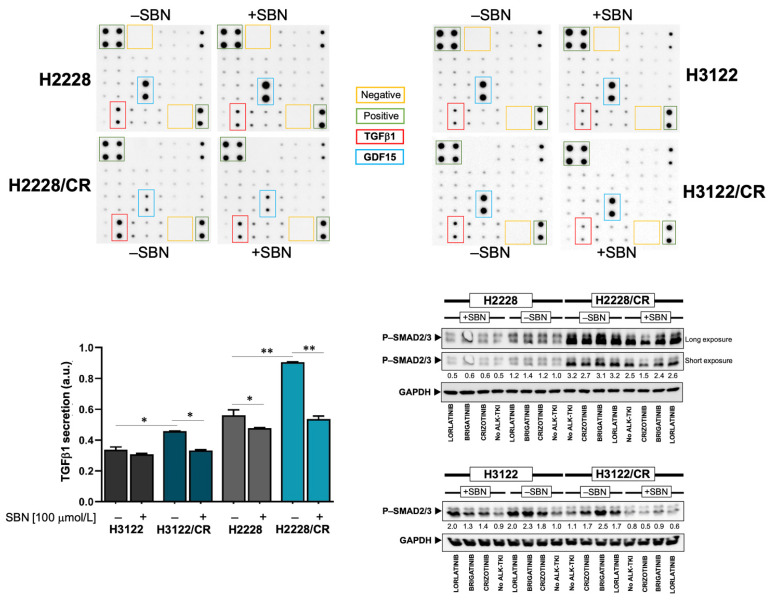
Effects of silibinin on the TGFβ/SMAD signaling axis in ALK–TKIs-resistant NSCLC cells. Top: Low-serum (0.2% FBS), 24-hour-conditioned media from H2228/H2228CR and H3122/H3122CR cells cultured in the absence or presence of silibinin (100 μmol/L) were assayed for the content of 25 TGFβ-related secreted proteins, as described in the Materials and Methods section. Shown are representative results (*n* = 3) revealing conspicuous changes in TGFβ1 and GDF15. Bottom left: The intensity of TGFβ1 dots was measured using the ImageJ software. Relative changes in TGFβ1 secretion were calculated following subtraction of membrane background signal and normalization to positive control readings. Bottom right: Expression levels of phospho-SMAD2/3 were detected by immunoblotting in H2228/H2228CR and H3122/H3122CR cells treated with crizotinib, brigatinib or lorlatinib (1 μmol/L, 24 h) in the absence/presence of silibinin (100 μmol/L) using a specific antibody. The intensity of the bands was measured using the ImageJ software. The fold-change of each protein relative to untreated samples was calculated using GAPDH as a loading control. The figure shows representative immunoblots of multiple (*n* = 3) independent experiments. The uncropped western blot figures were presented in Appendix A. * *p* < 0.05 and ** *p* < 0.005.

**Figure 6 cancers-14-06101-f006:**
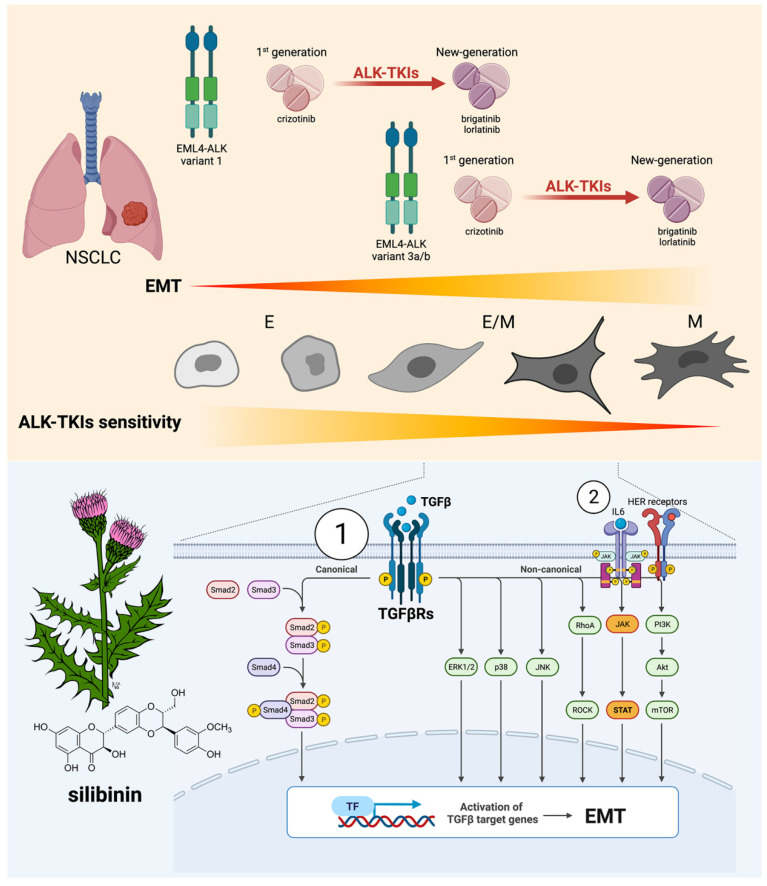
EMT-ness and resistance to multiple-generation ALK–TKIs: a therapeutic opportunity for silibinin. Top: The plasticity along the EMT spectrum might determine the propensity of ALK-rearranged NSCLC cells to exhibit cross-resistance to multiple-generation ALK–TKIs. The more epithelial an ALK-rearranged NSCLC cell population is (e.g., EML4-ALK variant 1), the lower the capacity to acquire a mesenchymal phenotype refractory to new-generation ALK–TKIs, and vice versa, the more mesenchymal an ALK-rearranged NSCLC cell population is (e.g., EML4-ALK variant 3a/b), the higher the capacity to acquire a mesenchymal phenotype refractory to new-generation ALK–TKIs. ALK-rearranged NSCLC cells gaining a bona fide mesenchymal phenotype caused by a late, full EMT upon chronic exposure to crizotinib, but not those acquiring only a partial/hybrid E/M transition state, exhibit an augmented resistance to the 2nd generation ALK–TKI brigatinib and complete refractoriness to the 3rd generation ALK–TKI lorlatinib. Bottom: The flavonolignan silibinin can overcome EMT-driven cross-resistance to new-generation ALK–TKIs by attenuating the hyperactivation of the TGFβ/SMAD signaling axis (**1**). Nonetheless, silibinin can exert additional ALK–TKIs sensitizing effects via direct inhibition of STAT3 [75] and EGFR [76] (**2**), thereby preventing a functional landscape of resistance to ALK inhibition in NSCLC involving the activation of the IL6/JAK1/STAT3 [77] and HER [78] signaling pathways. Created with Biorender.com.

## Data Availability

All data generated or analyzed during this study are included in this published article (and its Appendix A file).

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
