# Peer review of "Silibinin Overcomes EMT-Driven Lung Cancer Resistance to New-Generation ALK Inhibitors"

_cancers, 2022, doi:10.3390/cancers14246101_

Round 1

Reviewer 1 Report

In this manuscript, the authors add valuable further weight to the emerging model that epithelial-to-mesenchymal transition (EMT) represents an important off-target resistance mechanism for ALK+ lung cancer patients treated with targeted ALK inhibitors. They show that EMT is a common response to treatment with the first generation ALK TKI, crizotinib. Moreover, they provide evidence that silibinin, a natural compound known to interfere with EMT, might slow the development of resistance, potentially through inhibiting TGFbR signalling, leaving patients more responsive to second and third generation ALK inhibitors.

Overall, the manuscript is well written and referenced, and the data presented of high quality. Most of the study was performed with two ALK+ cancer patient cell lines, one expressing EML4-ALK variant 1 (H3122) and the other expressing EML4-ALK variant 3 (H2228). For each of these lines, a crizotinib-resistant (CR) derivative had been previously generated by chronic exposure to crizonitib. Hence, the bulk of experiments compared the properties and responses of the parental vs CR paired cell lines. This is a legitimate approach and the results are interesting. However, it is important to note that, although the CR cells did not have on-target ALK mutations, the parental lines already have different properties as a result of the different variant expressed and there are likely to be multiple mechanisms of resistance in the CR cells. These points should be considered in the Discussion.

The brightfield images in Fig 1A need to be improved as it is currently difficult to see the cells and the change in morphology quantified. The text associated with Fig 1 (including the figure title) should also be revised as the presence of EMT morphology and resistance to ALK TKIs isn’t of itself demonstration of a related response.

The sensitizing effect of silibinin to ALK inhibitors reported in Figure 2 is key to the conclusions drawn. It is therefore important that these data are quantified. However, while the MTT assay data in Fig 2A is quantified with good statistical significance, some form of quantification should be added for the colony formation assay in Fig 2B. I couldn’t find the reported Table 1.

The Western blot data in Figs 3A & B also need to be quantified. Indeed, based on the blots shown, I’m not convinced that vimentin is increased in the H2228/CR cells as claimed, while it’s not clear why b-actin levels are significantly increased in the H2228/CR compared to H2228 cells.

In several places, including the Conclusions, the authors claim that silibinin directly inhibits TGFbR kinase activity. However, this conclusion is not supported by data from the ALK+ cell lines. They have in silico evidence for binding to the catalytic site and some rather inconclusive experimental evidence based on a fluorophore displacement FRET assay for binding to TGFbR. I strongly recommend that Western blot assays are performed in the ALK+ lung cancer cells to analyse phospho-TGFbR and phospho-SMAD2/3. This would provide more persuasive evidence for kinase inhibition, although even with this data they should revise the text to be more circumspect about this being direct. Furthermore, I think that without these additional data, the bottom half of Figure 5 would be better removed as the direct inhibition of TGFbR activity by silibinin remains unproven.

The Reference list is missing reference 36.

Reviewer 2 Report

In this study, the authors described in a first part the consequence of EMT transition on resistance to first, second and third generations of ALK TKI. The study was performed on two resistant cell lines displaying different stage of EMT. In a second part of the study, the authors described the impact of a plant extract silibinin able to revert EMT and able in consequence to restore sensitivity to ALK inhibitors. Finally, the authors suggest that silibinin directly binds TGFb receptor and explain effect on EMT reversion and restoration of sensitivity to ALK TKI.

Although, these data are potentially interesting, they sound too preliminary, notably for the direct targeting of TGFbR by silibinin. Notably, only one experiment based on ATP displacement suggests an interaction between TGFbR and silibinin. The first part of the study about the impact of EMT on resistance is more complete. The manuscript could be focus on this aspect.

Specific comments:

Line 28: the term “born” sound inappropriate in this context.

Line 95: It should be mentioned that data of the literature demonstrate that Silibinin is able to downregulated TGFb signaling.

Figure 1A: Morphological changes mentioned in text is not visible in the Figure 1A. A zoom could be proposed. An actin staining and/or a labelling of the tight junction could evidence the phenotype.

Figure 2A: The authors propose that effect of silibinin plus TKI, observed in H3122 cells, is the consequence of silibinin toxicity in single treatment in these cells (line 299). However, silibinin condition alone is not shown in figure 1A and table S1. Such condition should be shown. In the same line, silibinin condition alone should be shown in figure 2B.

Figure 3A: The mention “EMT” in the figure should be remove, since it suggests that experimental condition should induce EMT

Figure 3: Effect of silibinin on SMAD phosphorylation and expression should be determined in H2228 and H3122 cells.

Figure 3: WB of P-SMAD could be quantified to assess decrease of it phosphorylation. The panel could be numbered (panel C).

Figure 4: IC50 of silibinin required to displace ATP (more than 50microM) and more generally the concentration of 100microM used in the previous experiments is quite high for an active and specific compound. At least, dose response should be performed.

Reviewer 3 Report

The authors have presented a comprehensive manuscript investigating regulatory role of silibinin of ALK–rearranged non–small-cell lung cancer (NSCLC) tumors. I have the foll0wing minor comments:

1. The authors should briefly describe the method to develop H2228/CR cells and evaluation method. How did the authors revalidate the resistance characteristics in the cells after freeze-thaw cycle. If space /word limit is an issue, the authors can summarize section 2.3 as it is a very well known technique.

2. section 2.10, reference are missing.

3.  Fig. 1B, statistical analysis is missing.

4. The authors should add a paragraph highlighting their expected outcome with invivo experiments.

Round 2

Reviewer 1 Report

The authors have responded very positively to my original review and I have no further requests. I am very happy to now recommend publication of this interesting manuscript.